# Potential of the Biomass of Plants Grown in Trace Element-Contaminated Soils under Mediterranean Climatic Conditions for Bioenergy Production

**María Pilar Bernal \*, Donatella Grippi and Rafael Clemente** 

Centro de Edafología y Biología Aplicada del Segura, CSIC, Campus Universitario de Espinardo, 30100 Murcia, Spain; d.grippi@um.es (D.G.); rclemente@cebas.csic.es (R.C.)
\* Correspondence: pbernal@cebas.csic.es

**Abstract:** Phytomanagement of trace element-contaminated soils combines sustainable soil remediation with the use of plant biomass for different applications. Consequently, phytostabilization using plant species useful for bioenergy production has recently received increasing attention. However, the water requirement of most of these species is a limitation for their use under Mediterranean climatic conditions. In this work, eight plant species growing naturally in mine soils contaminated by trace elements were evaluated for their use as bioenergy crops using thermochemical (combustion) and biochemical (anaerobic digestion) methods. The higher heating values of the biomass of the plants studied were all within a narrow range (16.03–18.75 MJ kg$^{-1}$), while their biochemical methane potentials ranged from 86.0 to 227.4 mL CH$_4$ (g VS)$^{-1}$. The anaerobic degradation was not influenced by the presence of trace elements in the plants, but the mineral content (mainly Na) negatively affected the potential thermal energy released by combustion (HHV). The highest annual energy yields from biogas or combustion could be obtained by the cultivation of *Phragmites australis* and *Arundo donax*, followed by *Piptatherum miliaceum*. Both options can be considered to be suitable final destinations for the biomass obtained in the phytostabilization of trace element-contaminated soils and may contribute to the implementation of these remediation techniques in Mediterranean areas.

**Keywords:** anaerobic digestion; biogas; combustion; Higher Heating Value; phytoremediation

## 1. Introduction

Mining is one of the industrial activities with the greatest environmental impact [1]. The mining district of Cartagena-La Unión (Murcia, southern Spain) was exploited intensely for ores containing silver, lead, zinc and other metallic minerals from ancient times until the definitive cessation of mining activities in 1990. As a consequence, the landscape with an area of about 50 km² was transformed and the old vegetation cover disappeared, along with habitat and numerous species [2]. The dispersion of mining waste into the surrounding environment has led to colonization by spontaneous plant species that show adaptation to soil metal contamination [3]. Recently, containment and remediation measures have been proposed for mine-affected areas [4]. Non-invasive remediation technologies, such as phytostabilization, can be effective for the in situ stabilization of metals in the contaminated sites of the Cartagena-La Unión mining district [5,6]. Through phytostabilization, the establishment of a self-sufficient vegetative cover reduces the risk of an uncontrolled transfer of pollutants into the surrounding environment. However, the use of the plant biomass resulting from remediation is still a challenge for the implementation of phytoremediation strategies [7].

The use of contaminated land for the production of valuable biomass (such as the production of timber, bioenergy crops, biofortified products, ecomaterials, etc.) is considered essential for the commercial success of the phytotechnologies and falls within the concept of phytomanagement [8,9]. Therefore, in phytomanagement, the remediation strategies

are combined with sustainable site management options, which results in a net gain (or at least no gross reduction) in soil function and ecosystem services, as well as effective risk management [10]. Thus, the use of potential energy crops in the phytoremediation of soils contaminated by trace elements (TEs) has a double objective: the natural remediation of the site and the production of renewable energy [9]. Up to now, energy crops have mainly comprised species with a high biomass per unit area when grown in good agricultural soils, but now special attention is being paid to species capable of producing reasonable yields when grown on marginal land [11], which may help to avoid competition for resources with food production [12].

The lignocellulosic biomass from woody crops and grasses can be a source of re-newable energy through thermochemical and biochemical conversion methods [13]. The physical and chemical characteristics of different plant species in relation to their use as biomass crops for bioenergy production, the so-called second-generation biofuels, have previously been reviewed [14]. The richness in certain organic components of the plants, especially cellulose, hemicellulose and lignin, defines their usefulness for the production of different biofuels, such as bioethanol, biogas, biomethane, biohydrogen, syngas or bio-oil, and for combustion to obtain renewable energy [11,15,16]. For dry biomass conversion processes such as combustion, pyrolysis or gasification, the main properties to be consid-ered are the moisture content, calorific value, ash content, alkali and metal content and cellulose/lignin ratio, while for wet biomass conversion (such as anaerobic digestion or alcoholic fermentation), the most relevant parameters are the moisture content and the cellulose/lignin ratio [11].

Perennial energy crops (*Miscanthus* spp., *Ricinus communis*, *Jatropha curcas*, *Populus* spp., *Salix* spp.) in short-rotation coppices and certain high-biomass annual crops have been eval-uated for bioenergy production in sustainable phytoremediation [9,17]. However, the high water requirements of such energy crops limit their cultivation in Mediterranean regions with water scarcity. In this sense, the use of native species from marginal or contaminated land, adapted to the climatic conditions of the area to be remediated, as bioenergy crops can comply with several objectives, such as: limiting the introduction of non-native species into the local soil ecosystem, promoting soil biological activity in such degraded soils, avoiding competition for agricultural soil with food crops and improving the $CO_2$ balance. Never-theless, research is needed to evaluate the potential of native species from arid or semiarid areas for bioenergy production. According to Grippi et al. [14], plant species belonging to the families Poaceae, Asteraceae and Brassicaceae have characteristics suitable for their use as biofuels. Previous studies have evaluated the potential for bioenergy production of sev-eral wild species used for the remediation of TEs-contaminated soils under Mediterranean conditions, such as milk thistle (*Silybum marianum*), *Dittrichia viscosa*, *Piptatherum miliaceum* and *Nicotiana glauca* [18–21]. Thus, the plants growing spontaneously in contaminated soils can be considered the best candidates for phytoremediation, and at the same time can be a source of biomass for bioenergy production, to develop phytomanagement strategies.

However, there is little information on the energetic potential of these species and how their composition, influenced by the soil pollution, will positively or negatively affect their use for the production of bioenergy. The plant species used in phytostabilization, in addition to their adaptation to the soil and climatic conditions of the area, must have low TEs concentrations in their aerial (harvestable) parts, with restricted root-to-shoot transport. We hypothesized that the low TEs accumulation in the aerial parts of the plants makes their biomass adequate for bioenergy production, as low concentrations of potentially toxic elements will not negatively affect the thermochemical and biological processes involved in the production of thermal energy, through combustion, and biogas, by means of anaerobic digestion.

Therefore, the objective of this study was to identify and characterize plant species that grow naturally in the Cartagena-La Unión mining district, regarding their potential for bioenergy production and the phytomanagement of TE-contaminated soils. For this, the potential for thermal energy and biogas production were studied under laboratory

conditions, and the results were related to plant composition to define phytomanagement strategies based on bioenergy production.

## 2. Materials and Methods

### 2.1. Plant Sampling and Chemical Analyses

Eight different plant species growing spontaneously in the areas of El Gorguel and El Llano del Beal, within the Sierra Minera of La Unión-Cartagena, were sampled by cutting the aerial part (shoots) of several individual fully developed plants. The area of study is characterized by an average annual temperature of 17.7 °C with an annual precipitation of <300 mm, with occasional torrential rainfall during late summer or autumn. The average concentrations of the main TEs of the soils for El Gorguel and El Llano del Beal, respectively (mg kg$^{-1}$), are: As 408–664, Cd 35–19, Cu 116–193, Pb 6360–10188, and Zn 12048–9686 [6,22].

The species sampled were, in El Gorguel: *Arundo donax* L. (Poaceae), *Phragmites australis* (Cav.) Trin. ex Steud. (Poaceae), *Piptatherum miliaceum* (L.) Coss. (Poaceae) and *Foeniculum vulgare* Mill. (Apiaceae); and in El Llano del Beal: *Dittrichia viscosa* (L.) Greuter (Asteraceae), *Zygophyllum fabago* L. (Zygophyllaceae), *Bituminaria bituminosa* L. Stirton (Fabaceae) and *Atriplex halimus* L. (Amaranthaceae). The fresh samples were washed with deionised water, dried at 60 °C for 48 h and ground to <0.5 mm in a stainless-steel laboratory mill (Cyckone Mill-Twister, Retsch, Haan, Germany) prior to experiments and analysis.

The elemental composition of the plant samples (P, K, micronutrient and TEs concentrations) was determined by inductively coupled plasma optical emission spectroscopy (ICP-OES; THERMO ICAP 6500 DUO instrument, Thermo Scientific), after microwave-assisted digestion (ETHOS1, Milestone) with $H_2O_2$ and $HNO_3$ (1:4 *v/v*). The analytical accuracy was verified with a certified reference material (NCS DC 73349). Total solids (TS) were determined by drying at 105 °C and volatile solids (VS) were determined by ashing at 550 °C in a muffle furnace (Carbolite AAF 11/3, Hope Valley, UK) according to EPA method 1684 [23]. The concentrations of lignin and holocellulose were determined according to the American National Standard method [24,25]. The elemental (ultimate) analysis of C, N, S and H was carried out using a LECO CHNS-932 analyzer. The concentration of soluble carbohydrates was determined with the anthrone method [26]. All the analyses were performed at least in duplicate.

The higher heating value (HHV; MJ kg$^{-1}$) of the plant biomass was calculated from the elemental analysis, using the method of Sheng and Azevedo [27], according to the following equation:

$$\text{HHV (MJ kg}^{-1}) = -1.3675 + 0.3137 \times \text{C\%} + 0.7009 \times \text{H\%} + 0.0318 \times \text{O\%} \tag{1}$$

### 2.2. Anaerobic Degradation

The potential for biogas production of the plants was determined through anaerobic digestion, using the ANKOM Gas Production System with 310 mL-capacity bottles (ANKOMRF, ANKOM Technology, Macedon, NY, USA). With this system, the increase in pressure caused by the generation of biogas in the absence of oxygen is automatically recorded. The anaerobic inoculum used was collected from an urban wastewater treatment plant under mesophilic conditions (reactor capacity 7612 m$^3$; hydraulic retention time 29.7 days) and incubated at 37 °C for 24 h. The composition of the inoculum was: pH 7.37 $\pm$ 0.03, electrical conductivity 14.78 $\pm$ 3.58 dS m$^{-1}$, TS 21.67 $\pm$ 4.13 g L$^{-1}$ and VS 14.02 $\pm$ 2.12 g L$^{-1}$.

Aliquots (0.5 g) of each plant species were mixed with 100 mL of pre-incubated inoculum in individual bottles (VS ratio 1:3, substrate:inoculum) and the corresponding mixtures were then incubated in the absence of oxygen at a temperature of 37 °C under continuous stirring [20]. The anaerobic conditions were obtained by flushing the headspace of each bottle with an $N_2$/$CO_2$ mixture (80:20, *v:v*). A control sample without plant material and a positive control containing 0.5 g of cellulose were also tested. The samples were run

in duplicate. The pressure generated inside the containers was recorded at 15 min intervals throughout the experiment. The biogas produced was expressed as the volume (mL) of biogas per unit (g) of VS of the plant. The volume of biogas produced was calculated using Avogadro's ideal gas law, as shown below:

$$\text{Biogas (mL g}^{-1}) = [\text{VH} \times (\text{Ps}-\text{Pi}) \times \text{R} \times \text{T} \times 22.414 \times 1000]/\text{m} \tag{2}$$

where VH is the volume of the headspace (L); Ps is the pressure in kPa of the sample; Pi is the pressure in kPa of the inoculum; R is the gas constant (8.314463 L Pa $K^{-1}$ $mol^{-1}$); T is the temperature in kelvin (K); and m is the weight of the sample (g of VS).

The experiment finished when the biogas production was almost stable, without any further increase (14–16 days). Then, biogas aliquots (in duplicate) from each bottle were taken through the septum port using a 10 mL glass syringe, and samples were kept in vacuum containers prior to analysis [28]. These gas samples were considered representative of the biogas composition of the gases that were generated and that accumulated in the bottles throughout the incubation period. The percentage of methane ($CH_4$) in the biogas samples was analyzed using a gas chromatography system (Agilent 490 Micro GC, Santa Clara, CA, USA).

The experimental results of the biogas production were described by a first-order kinetic model:

$$B_m = B_0 \times (1 - e^{-K \times t}) \tag{3}$$

where $B_m$ is the biogas in mL (g VS)$^{-1}$ produced at time t; $B_0$ indicates the maximum biogas production potential in mL (g VS)$^{-1}$; and *K* is the anaerobic degradation rate constant. The experimental results were fitted to a non-linear least square model (Marquardt–Levenberg algorithm) using the software Sigma Plot v. 14.0 (Systat Software Inc., San Jose, CA, USA).

The biochemical methane potential (BMP) of each plant material was calculated from the biogas production potential ($B_0$) and the percentage of $CH_4$. The theoretical BMP (TBMP) was calculated from the plant characteristics using a stoichiometric equation based on the concentrations of lignin, holocellulose, soluble carbohydrates, proteins (calculated by multiplying the N concentration by 6.25) and fats [29]:

$$\text{TBMP} = (1014 \times [\text{lipid}] + 496 \times [\text{protein}] + 415 \times [\text{carbohydrate}] + 727 \times [\text{lignin}]) \times 0.001 \tag{4}$$

The lipid concentration was estimated as 3.09 g kg$^{-1}$ dry matter [21,29] for the TBMP calculation (Equation (4)). The BMP/TBMP ratio indicates the anaerobic biodegradability.

### 2.3. Statistical Analysis

The effect of the plant species on the plant composition, HHV and biogas production was determined by a one-way ANOVA. Differences between means were determined using Tukey's test at $p < 0.05$. Before the statistical analysis, the data were tested for normality using the Kolmogorov–Smirnov test. Pearson's coefficients for the correlations between the degradation parameters and the TE concentrations in the plants were also determined (IBM SPSS Statistics 25 software).

## 3. Results and Discussion

### 3.1. Plant Composition and Calorific Value

The composition of the plants in terms of the main nutrients (N, P, K, Ca, Mg, Fe, Cu, Mn and Zn) depended on the plant species (Table S1). In fact, the greatest N concentration occurred in *Z. fabago* and the leguminous species *B. bituminosa*, while the highest K concentration occurred in the halophyte *A. halimus*, in line with Walker et al. [30].

The concentrations of TEs (As, Cd, Pb and Zn) were above the range considered as normal [31] in most of the plant species analyzed (Table 1). However, the plants collected did not show any evident toxicity symptoms and were all growing spontaneously in areas heavily affected by mining activity. In fact, most of the values did not reach the toxic limits for plants (As 5–20, Cd 5–30, Cu 20–100, Mn 400–1000, Pb 30–300, Zn 100-400;

all in mg kg$^{-1}$) [31], with the exception of Zn in most samples (in particular *D. viscosa*, *A. donax* and *P. australis*; Table 1) and As, Cd and Pb in *D. viscosa*, *P. australis* and *P. miliaceum*. However, such values are frequent in plants able to grow in mine soils [32]. The highest TE concentrations occurred for Al, As, Cd and Pb in *D. viscosa*, Cu, Mn and Zn in *P. australis* and Al in *P. miliaceum* (Table 1). Similar values for *D. viscosa* were found by Barbafieri et al. [33] and by Martínez-Fernández et al. [32]. However, Marchiol et al. [34] reported concentrations of Cd, Cu, Pb and Zn in *D. viscosa* (122, 55.9, 173 and 1172 mg kg$^{-1}$, respectively) higher than in the present experiment, and also Melendo et al. [35] found concentrations of Pb (307–2084 mg kg$^{-1}$) in plants of this species higher than in the present experiment. The concentration of As in *P. australis* was within the range of values found by Baroni et al. [36] and Álvarez-Robles et al. [22] for plants growing in mine soils. Likewise, the TEs concentrations in *P. miliaceum* plants were very similar to those found by Párraga-Aguado et al. [37], Arco-Lázaro et al. [38] and Clemente et al. [39] in soils from the same mining area. However, the concentrations of As and Pb in this species were lower than those previously reported [36,40], while the results for Cd and Cu were similar to those found in the aerial parts of this species by Kabas et al. [40] and Marchiol et al. [34]. The concentrations of TEs in *B. bituminosa* were within the range previously reported in soils from this mining area [38]. Contrastingly, the TEs concentrations in *A. halimus* and *Z. fabago* were much lower than those reported for plants grown in mine tailings with extremely high TE concentrations [41]. The plants of *F. vulgare* showed low TEs concentrations overall, while *A. donax* had quite high Zn concentrations (similar to those in *D. viscosa*), in agreement with the results of Barbafieri et al. [33]. It is interesting to note the high Na concentrations in *Z. fabago* and *A. halimus* (18.1 and 10.3 g kg$^{-1}$; Table S1, Supporting Information), due to their halophytic character and Na$^+$ accumulation capacity [30,42]. Significant positive correlations were found between the concentrations of the different TEs in the plants (Table S2), which is a common feature of plants growing in metalliferous and TE-contaminated soils [39,41,43].

The content of VS in the biomass is a parameter of great relevance regarding the possible use of the different species for bioenergy production. It indicates the organic matter of the biomass that can be combusted or potentially biodegraded to produce biogas. The VS concentrations were high for all species (80.6–93.7%; Table 2), close to the values found for other plant species such as *P. miliaceum*, *S. marianum*, *Helianthus annuus* L. and *N. glauca* (77.7–87.2%) [20]. The ash content is also a relevant factor for the production of energy from biomass, as it quantifies the mineral components. The results obtained were similar to those observed for typical biomass materials, such as *Miscanthus*, wheat or barley straw (2.8–6%) [11]. High ash content can lower the available energy released in a thermochemical process, and is especially relevant for combustion [11].

**Table 1.** Trace element concentrations (mg kg$^{-1}$) in the aerial part of the plants used in the experiment (average ± se; n = 2).

| Plants | Al | As | Cd | Cu | Fe | Mn | Pb | Zn |
|---|---|---|---|---|---|---|---|---|
| *D. viscosa* | 563.4 ± 47.0 a | 5.6 ± 0.08 a | 8.0 ± 0.31 a | 13.70 ± 0.12 b | 1099.4 ± 29.3 a | 70.9 ± 0.2 c | 108.5 ± 3.4 a | 347.5 ± 2.9 b |
| *A. halimus* | 76.4 ± 4.7 d | 0.7 ± 0.15 c | 2.8 ± 0.03 d | 5.81 ± 0.17 e | 106.1 ± 6.1 fg | 73.5 ± 0.9 c | 11.1 ± 0.8 e | 254.9 ± 6.1 c |
| *B. bituminosa* | 200.0 ± 1.5 c | 1.4 ± 0.10 bc | 2.2 ± 0.07 d | 7.46 ± 0.05 d | 282.4 ± 4.1 d | 70.6 ± 2.0 c | 45.4 ± 1.7 c | 197.7 ± 3.1 d |
| *Z. fabago* | 88.6 ± 14.6 d | 1.9 ± 0.26 b | 0.5 ± 0.1 e | 8.00 ± 0.18 d | 101.6 ± 1.4 g | 52.5 ± 0.1 e | 5.1 ± 0.3 g | 158.2 ± 1.4 e |
| *A. donax* | 364.2 ± 6.9 b | 0.6 ± 0.12 c | 4.1 ± 0.06 c | 11.19 ± 0.21 c | 230.3 ± 4.0 e | 84.2 ± 1.4 b | 17.8 ± 0.7 d | 314.8 ± 3.8 b |
| *P. australis* | 336.9 ± 4.2 b | 3.0 ± 0.27 ab | 5.7 ± 0.16 b | 17.74 ± 0.24 a | 446.2 ± 9.9 c | 241.4 ± 0.4 a | 77.1 ± 3.4 b | 646.0 ± 19.3 a |
| *P. miliaceum* | 571.7 ± 22.5 a | 2.1 ± 0.74 b | 5.4 ± 0.05 b | 14.46 ± 0.27 b | 670.0 ± 8.2 b | 59.2 ± 0.5 d | 94.1 ± 1.9 ab | 276.8 ± 1.7 c |
| *F. vulgare* | 95.7 ± 3. 8 cd | 1.3 ± 0.03 bc | 0.56 ± 0.02 e | 5.56 ± 0.1 e | 121.2 ± 3.0 f | 34.4 ± 0.4 f | 8.3 ± 0.3 f | 76.3 ± 2.5 f |
| ANOVA | *** | *** | *** | *** | *** | *** | *** | *** |

***: significant at $p < 0.001$. Values followed by the same letter in each column do not differ significantly according to Tukey's test at $p < 0.05$.

**Table 2.** Chemical composition of the plants used in the experiments (average values ± se, n = 2).

| Plants | Ash (%) | VS (%) | Lignin (%) | Holocellulose (%) | Soluble Carbohydrates (g kg$^{-1}$ dw) |
|---|---|---|---|---|---|
| *D. viscosa* | 7.03 ± 0.02 c | 90.82 ± 0.16 b | 26.96 ± 0.72 a | 50.94 ± 7.21 bc | 54.85 ± 2.95 bc |
| *A. halimus* | 14.25 ± 0.14 b | 84.14 ± 0.05 c | 22.40 ± 0.23 b | 62.11 ± 5.73 abc | 12.40 ± 0.40 f |
| *B. bituminosa* | 5.31 ± 0.06 c | 93.19 ± 0.02 a | 20.09 ± 0.45 bc | 65.19 ± 0.57 abc | 26.5 ± 0.80 ef |
| *Z. fabago* | 16.28 ± 0.31 a | 80.57 ± 0.57 d | 17.08 ± 0.92 c | 48.58 ± 6.07 c | 35.15 ± 0.95 de |
| *A. donax* | 6.26 ± 0.19 c | 93.74 ± 0.19 a | 20.92 ± 1.67 bc | 68.23 ± 0.59 abc | 83.85 ± 3.15 a |
| *P. australis* | 6.81 ± 0.45 c | 93.19 ± 0.45 a | 20.87 ± 0.55 bc | 72.32 ± 056 a | 49.5 ± 0.01 cd |
| *P. miliaceum* | 7.09 ± 0.61 c | 92.92 ± 0.60 a | 18.42 ± 0.33 bc | 72.42 ± 0.65 a | 44.15 ± 3.05 cde |
| *F. vulgare* | 6.92 ± 0.33 c | 93.08 ± 0.33 a | 19.59 ± 0.33 bc | 70.98 ± 0.38 ab | 69.35 ± 7.35 ab |
| ANOVA | *** | *** | ** | ** | *** |

** and ***: significant at $p < 0.01$ and 0.001, respectively. Values followed by the same letter in each column do not differ significantly according to Tukey's test at $p < 0.05$.

Lignin is one of the main constituents of plants, and usually represents 15–35% of plant biomass, depending on the species. In the present experiment, the highest lignin concentrations were found in *D. viscosa*, while the other species did not show significant differences among them (Table 2). In general, the values obtained were higher than those of plant species used for bioenergy production, such as *Sorghum vulgare* (stalk) 18.69%, *Arundo donax* (stems) 16.8%, *Miscanthus sinensis* var. Giganteus (stems) 12.8%, *Panicum virgatum* (whole plant) 17.6%, wheat straw 15–20%, or switchgrass 5–20% [11,44–46]. The values for *A. donax* were within the range (19.2–24.3%) reported by Corno et al. [47]. The lignin content of each species is of great relevance for the use of plant biomass for energy purposes: high values can imply suitability for combustion, but may indicate low degradability for biogas production [14].

The concentration of soluble carbohydrates was highest in *A. donax* and *F. vulgare* (Table 2), which indicates the feasibility of their use for anaerobic digestion for biogas production or for alcoholic fermentation. However, the concentration of holocellulose (the sum of cellulose and hemicellulose) in all samples was within a narrow range, being slightly higher for *P. miliaceum* and *P. australis*. The holocellulose values found for *A. donax* were towards the upper limit of the range reported by Corno et al. [47], but similar to those found by Cencič et al. [48] for this species. The values for the rest of the species were generally higher (except for *D. viscosa* and *Z. fabago*) than those found in other lignocellulosic species (53–65%) [14].

The elemental (C, N, S, H and O) composition of the plants (Table 3) was used to determine their HHV and lower heating value (LHV). The HHV represents the maximum amount of energy potentially recoverable from a biomass, which includes the latent heat of water vapor, while LHV takes into account that the latent heat contained in the water vapor is not always recoverable. The values of HHV were found to be within a rather narrow range (15.66–18.75 MJ kg$^{-1}$), the lowest values being found for *Z. fabago* and the highest for *A. donax* (Table 3). These values were close to the upper limit of the range found by Bernal et al. [20] for *P. miliaceum*, *S. marianum*, *H. annuus* and *N. glauca* (14.76–17.45 MJ kg$^{-1}$), and slightly higher than those determined for *S. marianum* and *H. annuus* plants using a bomb calorimeter (12.5–17.5 MJ kg$^{-1}$) [21]. The values reported by McKendry [11]—for different types of plants (fir 21 MJ kg$^{-1}$, Danish pine 21.2 MJ kg$^{-1}$, willow 20 MJ kg$^{-1}$, poplar 18.5 MJ kg$^{-1}$, cereal straw 17.3 MJ kg$^{-1}$, Switchgrass 17.4 MJ kg$^{-1}$, *Miscanthus* 18.5 MJ kg$^{-1}$)—and those of Boundy et al. [49], for herbaceous species (17.2 MJ kg$^{-1}$), are close to most of the values found here. The HHVs found were lower than that of coal (22.7 MJ kg$^{-1}$), but close to the values of forest residues and farmed trees (15.4 and 19.5 MJ kg$^{-1}$, respectively) [49].

Even though the lignin content of lignocellulosic biomass has been generally related to the heating value [50], no significant correlation was found between the lignin concentration and HHV in the present samples. However, the contents of VS and carbohydrates showed significant positive correlations with HHVs in the plants studied (r = 0.946 and 0.731, $p < 0.001$ and 0.05, respectively). The calorific value was not influenced by the presence of TEs in the plant biomass, as there were no significant correlations between the As, Cd, Cu, Pb or Zn concentration and HHV for the samples (data not shown). In fact, the Mg and Na concentrations in the plants correlated negatively with HHV (r = −0.765 and −0.914, $p < 0.05$ and 0.01, respectively), indicating that the salt accumulation in the plants can affect their calorific potential and therefore their use for combustion [11]. High alkali metal contents (Na, K, Mg and Ca) are important for any thermochemical conversion process [11], and in the present work, the lowest HHV values, in *A. halimus* and *Z. fabago*, coincided with the greatest total concentrations of those alkaline metals (sum of Na, K, Mg and Ca; Table S1). In fact, Walker et al. [30] evaluated the use of *A. halimus* plants for combustion and obtained 17.2 and 12.8 MJ kg$^{-1}$ for stems and leaves, respectively, in agreement with the results found here, but they considered that its energetic use should be limited to plants from non-saline soils. High ash contents may also lead to the problem of fouling and slagging in combustion equipment [20]. This is particularly relevant when the silica content

of the biomass is high and, especially, when soil particles remain on the plant material after harvesting [11].

*3.2. Anaerobic Digestion*

The volume of biogas produced after 16 days of anaerobic digestion ($B_m$) was lowest for *A. halimus*, followed by *Z. fabago* and *D. viscosa*, without significant differences among the rest of the species (*P. miliaceum*, *A. donax*, *P. australis*, *B. bituminosa* and *F. vulgare*; Table 4). In general, the values found were lower than those reported for other energy crops such as sunflower (up to 454 mL $g^{-1}$ VS) [51]. In addition, a lag-phase was found for *P. miliaceum* at the beginning of the experiment (Figure 1), indicating an initial difficulty for the microbial degradation. The composition of the plants cannot explain this behavior and, in fact, the $B_m$ value was highest for this species, while similar results ($B_m$ 235–270 mL (g VS)$^{-1}$) were found by Bernal et al. [20]. So, the initial lag-phase of microbial degradation did not limit the anaerobic process of biogas production.

The results fitted a first-order kinetic model at a high significance level (at $p < 0.001$ for all samples; data not shown). The values of $B_0$ mirrored those of $B_m$ and followed a very similar order for the different species studied: they were highest for *A. donax* and *P. australis*, without significant differences from the values for *B. bituminosa*, *P. miliaceum* and *F. vulgare*, and were lowest for *A. halimus* (Table 4). Therefore, the anaerobic degradation of the plant biomass was almost complete at the end of the experimental time. The highest value of the rate constant (*K*) was found for *Z. fabago* (Table 4), which indicates that the maximum production of biogas was reached earlier for this biomass than for the other plant species, regardless of the total amount of biogas produced. This can be attributed to the lower lignin concentration and C/N ratio of *Z. fabago* compared to the other species (Tables 3 and 4), as it has been previously shown that difficulty in the degradation of highly lignocellulosic materials leads to limitations during the hydrolysis phase [20,21]. The values of both $B_m$ and $B_0$ in the studied species were generally lower than those reported for energy crops (per g VS) such as maize (345 mL $g^{-1}$, whole plant), and also for other herbaceous species, such as nettle (210–420 mL $g^{-1}$), ryegrass (360 mL $g^{-1}$) and sunflower (454 mL $g^{-1}$, whole plant) [51–53]. Indeed, only the values of $B_0$ for *P. miliaceum* and *A. donax* were close to those found by Bernal et al. [20] for different species from phytoremediation experiments.

The variability in the $B_0$ and BMP values of crops may be due to differences in the lignin content [54,55] and to the physiological state of the plants at the time of the harvest. The biogas yield (and BMP) is decreased in late harvests [52]. Additionally, $B_0$ depends on the contents of total soluble carbohydrates, N and ash and on the ratio of lignin to ADF (acid detergent fiber) [54]. In fact, the high value of $B_0$ in *P. miliaceum* may be linked to its low lignin concentration and high holocellulose to lignin ratio. The BMP values were high for *A. donax*, *P. australis* and *P. miliaceum* (Table 4), with values close to those found for solid wastes derived from *H. annuus* oil extraction (between 107 and 227 mL $g^{-1}$) [56] and for maize, thistle and sorghum silage (267, 308, and 241 mL $g^{-1}$, respectively) [57], all expressed per g VS. However, the values obtained for the aerial vegetative parts of *S. marianum* (174 mL $g^{-1}$) and *H. annuus* (119 mL $g^{-1}$) by Hunce et al. [21] are close to the lowest values obtained in this study. Only the BMP values for *A. donax*, *P. australis* and *P. miliaceum* were in the range found by Gunaseelan [58] for *Morus indica* leaves, fruits and roots (332–461 mL $g^{-1}$). The values for *B. bituminosa*, *Z. fabago* and *D. viscosa* were close to those of *Jatropha curcus* leaves (an average of 230 mL $g^{-1}$), while lower values were obtained for *A. halimus* biomass. The values for *A. donax* were higher than those obtained for the same species in a laboratory reactor (130–150 mL $g^{-1}$) after 50 days at different inoculum ratios [59]. With the exception of *A. halimus*, the results are comparable to those from well-known energy crops such as *Miscanthus* and switchgrass (152–212 mL $g^{-1}$) [59].

**Table 3.** Elemental analysis, higher heating value (HHV), lower heating value (LHV) and C/N ratio of the plants (average values $\pm$ se; n = 2).

| Plants | C (%) | N (%) | S (%) | H (%) | O (%) | HHV (MJ kg$^{-1}$) | LHV (MJ kg$^{-1}$) | C/N |
|---|---|---|---|---|---|---|---|---|
| *D. viscosa* | 46.24 $\pm$ 0.09 a | 0.57 $\pm$ 0.01 e | 0.10 $\pm$ 0.02 d | 5.12 $\pm$ 0.08 d | 38.78 $\pm$ 0.03 c | 17.96 $\pm$ 0.03 c | 16.83 $\pm$ 0.01 b | 80.85 $\pm$ 1.44 a |
| *A. halimus* | 41.33 $\pm$ 0.01 e | 1.21 $\pm$ 0.00 b | 0.14 $\pm$ 0.02 cd | 4.66 $\pm$ 0.07 e | 36.81 $\pm$ 0.12 d | 16.03 $\pm$ 0.04 e | 15.01 $\pm$ 0.04 d | 34.27 $\pm$ 0.07 d |
| *B. bituminosa* | 45.01 $\pm$ 0.09 b | 1.48 $\pm$ 0.01 ab | 0.07 $\pm$ 0.01 d | 5.07 $\pm$ 0.02 d | 41.57 $\pm$ 0.14 b | 17.62 $\pm$ 0.04 d | 16.51 $\pm$ 0.05 c | 30.49 $\pm$ 0.26 d |
| *Z. fabago* | 40.33 $\pm$ 0.02 f | 1.99 $\pm$ 0.08 a | 0.52 $\pm$ 0.07 a | 4.75 $\pm$ 0.02 e | 32.99 $\pm$ 0.22 e | 15.66 $\pm$ 0.01 f | 14.62 $\pm$ 0.01 e | 20.35 $\pm$ 0.86 e |
| *A. donax* | 43.13 $\pm$ 0.02 c | 0.72 $\pm$ 0.02 cd | 0.32 $\pm$ 0.02 b | 7.50 $\pm$ 0.01 a | 42.07 $\pm$ 0.05 b | 18.75 $\pm$ 0.00 a | 17.11 $\pm$ 0.01 a | 59.93 $\pm$ 1.22 bc |
| *P. australis* | 43.38 $\pm$ 0.05 c | 0.59 $\pm$ 0.00 e | 0.29 $\pm$ 0.02 bc | 7.02 $\pm$ 0.01 b | 41.91 $\pm$ 0.09 b | 18.49 $\pm$ 0.01 b | 16.95 $\pm$ 0.01 ab | 73.84 $\pm$ 0.27 a |
| *P. miliaceum* | 42.70 $\pm$ 0.04 d | 0.66 $\pm$ 0.03 d | 0.19 $\pm$ 0.00 bcd | 6.38 $\pm$ 0.03 c | 42.98 $\pm$ 0.14 a | 17.87 $\pm$ 0.03 c | 16.46 $\pm$ 0.03 c | 64.43 $\pm$ 2.75 b |
| *F. vulgare* | 43.44 $\pm$ 0.11 c | 0.77 $\pm$ 0.02 c | 0.08 $\pm$ 0.02 d | 6.11 $\pm$ 0.07 c | 42.68 $\pm$ 0.25 a | 17.89 $\pm$ 0.08 c | 16.55 $\pm$ 0.09 c | 56.38 $\pm$ 1.39 c |
| **ANOVA** | *** | *** | *** | *** | *** | *** | *** | *** |

***: significant at $p < 0.001$. Values followed by the same letter in each column do not differ significantly according to Tukey's test at $p < 0.05$.

**Table 4.** Results of the anaerobic degradation of the plants: the parameters of the first-order kinetic model, biochemical methane potential (BMP), theoretical biochemical methane potential (TBMP) and anaerobic biodegradability (mean $\pm$ se; n = 2).

| Plants | $B_m$ (mL (g SV)$^{-1}$) | $B_0$ (mL (g SV)$^{-1}$) | $K$ (h$^{-1}$) | BMP (mL CH$_4$ (g VS)$^{-1}$) | TBMP (mL CH$_4$ (g VS)$^{-1}$) | Anaerobic Biodegradability (%) |
|---|---|---|---|---|---|---|
| *D. viscosa* | 180.9 $\pm$ 3.6 bc | 182.2 $\pm$ 5.0 bc | 0.012 $\pm$ 0.001 bc | 131.2 $\pm$ 3.6 bc | 497.9 $\pm$ 26.6 | 26.4 $\pm$ 0.7 bcd |
| *A. halimus* | 116.6 $\pm$ 10.7 c | 119.9 $\pm$ 14.8 c | 0.012 $\pm$ 0.004 bc | 86.0 $\pm$ 10.4 c | 576.5 $\pm$ 30.6 | 15.1 $\pm$ 2.9 c |
| *B. bituminosa* | 224.4 $\pm$ 3.2 ab | 239.2 $\pm$ 0.6 ab | 0.011 $\pm$ 0.001 bc | 177.6 $\pm$ 1.1 ab | 525.3 $\pm$ 6.2 | 33.8 $\pm$ 0.2 abc |
| *Z. fabago* | 173.8 $\pm$ 2.5 bc | 184.0 $\pm$ 0.1 bc | 0.025 $\pm$ 0.001 a | 130.6 $\pm$ 0.04 bc | 514.7 $\pm$ 39.9 | 25.5 $\pm$ 2.0 cd |
| *A. donax* | 261.4 $\pm$ 1.5 a | 315.9 $\pm$ 25.5 a | 0.006 $\pm$ 0.001 bc | 227.4 $\pm$ 18.4 a | 517.1 $\pm$ 9.8 | 43.9 $\pm$ 2.7 a |
| *P. australis* | 234.5 $\pm$ 9.6 ab | 307.0 $\pm$ 23.6 a | 0.004 $\pm$ 0.001 c | 218.0 $\pm$ 16.8 a | 533.5 $\pm$ 9.3 | 40.8 $\pm$ 2.4 ab |
| *P. miliaceum* | 277.5 $\pm$ 26.9 a | 292.3 $\pm$ 38.8 ab | 0.015 $\pm$ 0.003 ab | 204.6 $\pm$ 27.2 ab | 518.9 $\pm$ 2.7 | 39.5 $\pm$ 5.5 abc |
| *F. vulgare* | 211.5 $\pm$ 16.1 ab | 219.4 $\pm$ 15.3 abc | 0.008 $\pm$ 0.0002 bc | 154.6 $\pm$ 108 abc | 524.3 $\pm$ 2.0 | 29.5 $\pm$ 2.2 abcd |
| **ANOVA** | *** | ** | ** | ** | NS | ** |

** and ***: significant at $p < 0.01$ and 0.001, respectively. NS = not significant. Values followed by the same letter in each column do not differ significantly according to Tukey's test at $p < 0.05$.

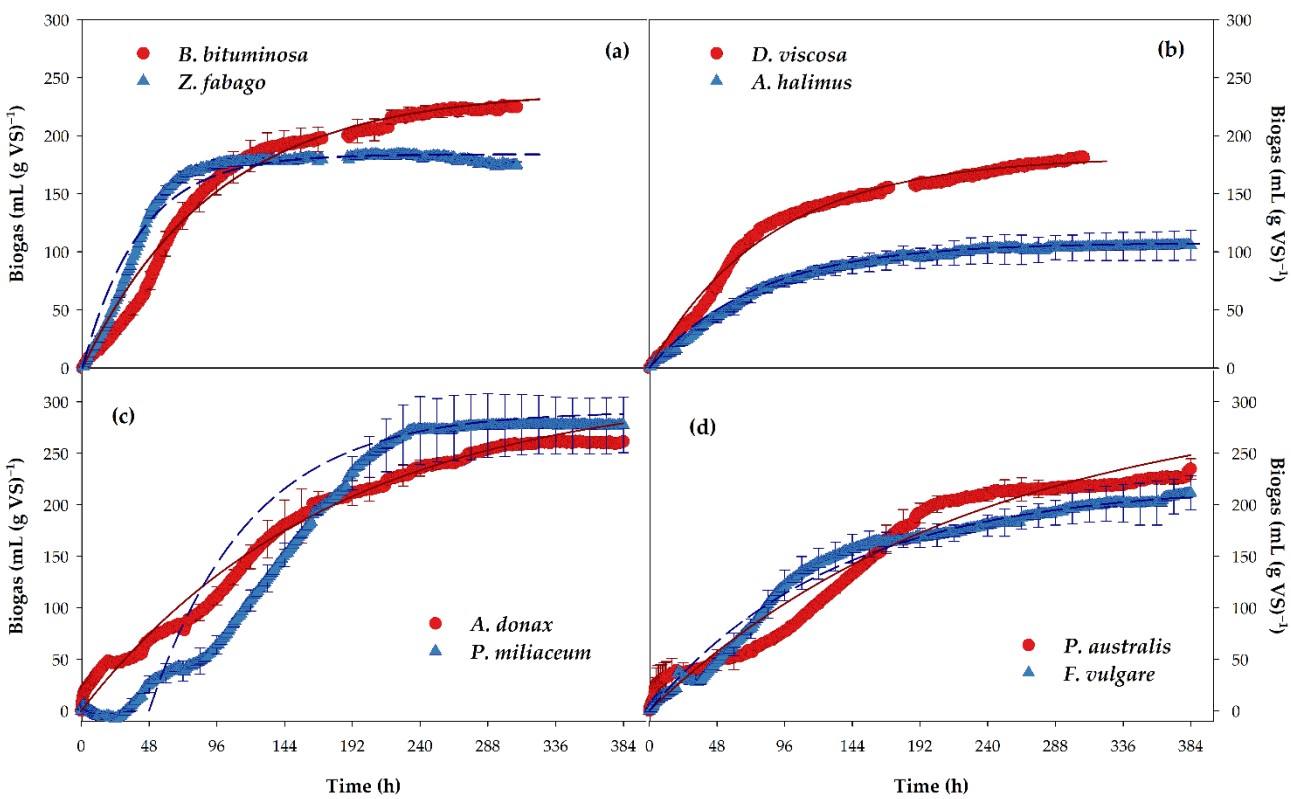

**Figure 1.** Biogas production from the plants (*Bituminaria bituminosa* and *Zygophyllum fabago* (**a**), *Dittrichia viscosa* and *Atriplex halimus* (**b**), *Arundo donax* and *Piptatherum miliaceum* (**c**) and *Phragmites australis* and *Foeniculum vulgare* (**d**)) under anaerobic conditions. The symbols represent the experimental data and the lines signify the degradation predicted by a first-order kinetic model for each sample (n = 2).

The TBMP had values between 497.9 (*D. viscosa*) and 576.5 mL g$^{-1}$ (*A. halimus*), without significant differences among the plant species. These values result in average anaerobic biodegradabilities between 15.1 (*A. halimus*) and 43.9% (*A. donax*). Only the values obtained for *A. halimus*, *Z. fabago*, *D. viscosa* and *F. vulgare* (Table 4) were lower than those found for wood, hedge and wild plants (32.7–44.9%) and for crops [29]. The low degradability found for *A. halimus* and *D. viscosa* may be associated with their higher lignin content compared to the other species (22.4 and 27.0%, respectively). It is known that lignin influences the production of biogas in the anaerobic digestion of lignocellulosic substrates such as agricultural crops, due to its low biodegradability associated with its complex structure [60]. Other factors, such as the salt content, may have reduced their anaerobic degradability, especially in *A. halimus* plants, which had a high Na concentration in their biomass. As with other halophytes, *A. halimus* accumulates Na$^+$ ions in its tissues for osmotic adjustment, as a tolerance mechanism in response to soil salinity [61]. In our experiment, negative correlations were found between the concentrations of Mg and Na and the BMP and anaerobic biodegradability (for Mg: r = −0.926 and −0.878, both at $p < 0.01$, respectively; for Na: r = −0.715 and −0.707, at $p < 0.05$, respectively). Contrastingly, K correlated positively with the P, Ca and Na concentrations in the plants (r = 0.832, 0.809 and 0.800, respectively, all $p < 0.05$), indicating a faster degradation of the plants with the highest concentrations of these elements.

The concentrations of VS, H and HHV correlated positively with the $B_m$, $B_0$ and BMP values in the plants (Table 5). However, no significant correlations were found between the anaerobic digestion parameters and the holocellulose, lignin or carbohydrates concentrations in the plants, which indicates that other factors influenced the anaerobic

degradation, such as the mineral content (Mg and Na) previously discussed, which also affected the thermal energy released by combustion (HHV).

**Table 5.** Pearson's coefficients for the correlations between plant composition, HHV and parameters related to the anaerobic digestion (n = 8).

| | Ashes | VS | N | H | HHV | $B_m$ | $B_0$ | *K* | BMP |
|---|---|---|---|---|---|---|---|---|---|
| $B_m$ | −0.721 * | 0.759 * | −0.448 | 0.789 * | 0.760 * | | | | |
| $B_0$ | −0.675 | 0.730 * | −0.471 | 0.896 ** | 0.792 * | 0.947 *** | | | |
| *K* | 0.721 * | −0.777 * | 0.739 * | −0.658 | −0.792 * | −0.336 | −0.476 | | |
| BMP | −0.693 | 0.740 * | −0.451 | 0.882 ** | 0.798 * | 0.943 *** | 0.998 *** | −0.488 | |
| Anaerobic biodegradability | −0.700 | 0.738 * | −0.451 | 0.869 ** | 0.016 | 0.955 *** | 0.994 *** | −0.451 | 0.996 *** |

*, ** and ***: significant at $p < 0.05$, 0.01 and 0.001, respectively.

Although a high lignin concentration in the plants may be indicative of low biodegradability [20,55], the species studied in the present experiment had lignin concentrations within a rather narrow range (17–27%), meaning that this parameter was not significantly related to any of the calorific or biogas parameters determined. Other parameters—such as the VS, Mg and Na concentrations in the plants and the carbohydrates content (for HHV)—seemed to be of greater relevance regarding the potential energetic transformation of the biomass of the different plant species studied.

The results obtained from the correlation tests show that the anaerobic degradation was not influenced by the presence of TEs in the plants (there were no significant correlations between the anaerobic digestion parameters and the TE concentrations in the plants; data not shown). Although Pb accumulation negatively affected the anaerobic degradation of *N. glauca* biomass (225–231 mg Pb kg$^{-1}$ dw), and therefore the biogas production [20], the concentration of Pb in the studied plants did not reach levels sufficient to negatively affect the anaerobic degradation of the plant biomass. All microbial processes require nutrients and certain TEs for their development. Some are essential macro- and micronutrients (N, P, S, K, Mg, Na, Ca, Fe, Cr, Co, Cu, Mn, Mo, Ni, Se, V and Zn), which are needed for enzymatic activities and other physiological plant processes [62]. Some TEs have shown positive effects on anaerobic digestion [63]; however, above certain concentrations, they become inhibitory or toxic [64]. Bozym et al. [65] reported the TE concentrations in plants that can be considered toxic for anaerobic digestion: Cd 180, Cu 40, Zn 100 and Pb 30 mg kg$^{-1}$. The concentrations of TEs determined in the plants were all lower than those values, except for Zn (Table 1). The contribution of certain micronutrients, such as Fe and Mn, could have had positive effects on the growth of almost all types of microorganism in the anaerobic digesters [66].

*3.3. Potential Energy Yield*

The potential for energy production (thermal and biogas) from the biomass of the different plant species collected at the contaminated site was evaluated using the HHV and BMP results (Tables 3 and 4). The potential thermal energy was calculated from the LHV, which considers the heat of condensation of the water vapor formed (calculated based on the H concentration) in the ultimate analysis (Table 3). The potential energy from anaerobic digestion was calculated from the BMP values and considering an HHV of 39.8 MJ m$^{-3}$ for pure $CH_4$. The annual yield of the plants depends on the cultivation conditions, but for each plant species under field conditions, the yield data considered were (Mg ha$^{-1}$ of dry matter): *A. halimus* 10; *B. bituminosa* 5; *A. donax* 33; *P. australis* 32; *P. miliaceum* 15 and *F. vulgare* 6.6 [30,47,67–70]. Information on the annual yields of *D. viscosa* and *Z. fabago* is not available, so an annual biomass production of 10 Mg ha$^{-1}$ of dry matter, as for *A. halimus*, was estimated.

In all cases, the potential energy from the combustion of the dry biomass was greater than that from its anaerobic digestion for biogas production (Figure 2). The highest values of the potential energy from biogas could be obtained by the cultivation of the highly

productive species *P. australis* and *A. donax*, with 298 and 319 GJ ha$^{-1}$ y$^{-1}$, respectively, followed by *P. miliaceum* with less than half that value (131 GJ ha$^{-1}$ y$^{-1}$). The rest of the species had much lower values, within a narrow range. The annual biomethane yields of *A. donax* and *P. australis* (8005 and 7486 m$^3$ ha$^{-1}$ y$^{-1}$, respectively) are similar to the lowest value of 9580 m$^3$ ha$^{-1}$ y$^{-1}$ reported by Corno et al. [47], although the values can vary significantly according to the environmental and agronomic conditions [47]. Lower potential energy values for *A. donax* (113–130 GJ ha$^{-1}$ y$^{-1}$) were reported by Yang and Li [59], due to a lower annual yield of the crop and partial degradation of the main components (cellulose, hemicellulose and proteins) during the study.

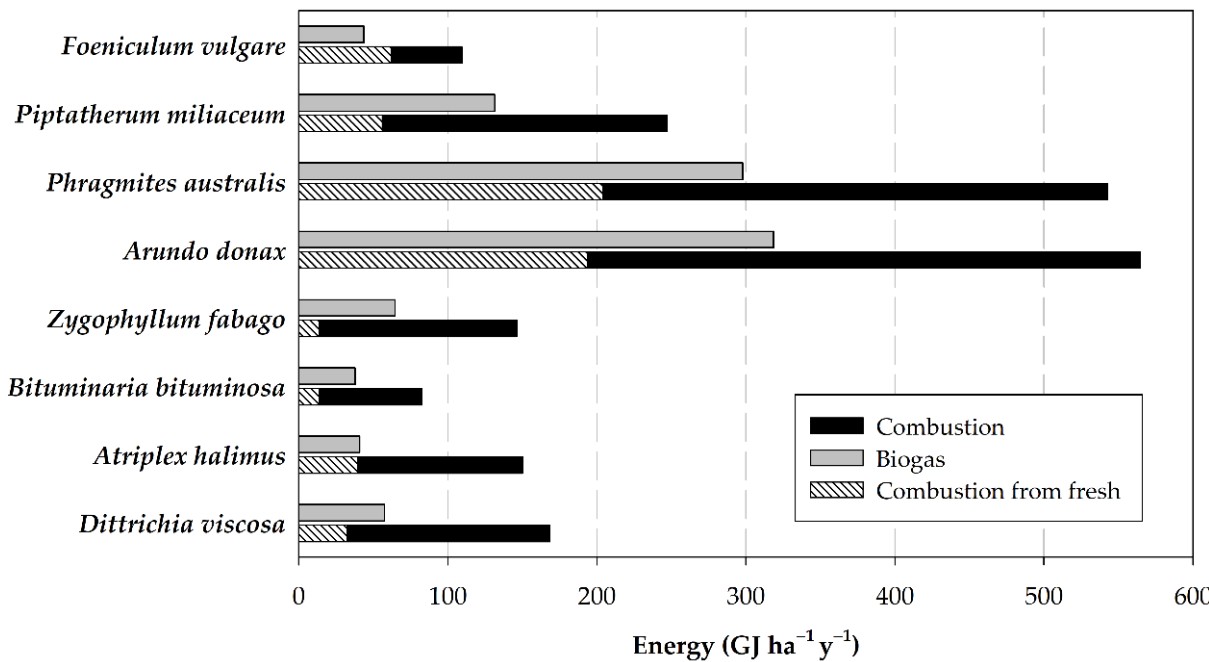

**Figure 2.** Potential energy production per hectare of land of the different plant species, by combustion (according to LHV) and by biogas production through anaerobic digestion. Comparison of the energy generated by combustion from the dry biomass (solid) and from the fresh biomass (dashed).

The potential annual thermal energy yield (combustion) could reach 542 and 564 GJ per hectare with *P. australis* and *A. donax*, respectively, due to their high annual biomass yield (Figure 2). The species least suitable for combustion would be *B. bituminosa*, followed by *F. vulgare*, while *P. miliaceum* could have good efficiency for thermal energy production (247 GJ ha$^{-1}$ y$^{-1}$). McKendry [11] reviewed the energy yields for certain biomasses, which showed the following order (GJ ha$^{-1}$): wheat straw 123, switchgrass 139, poplar 173–259, short-rotation coppice willow 187–280 and *Miscanthus* 222–555 (depending on the annual crop yield). According to these values, *P. australis* and *A. donax* could be considered comparable with the well-known high-biomass energy crop *Miscanthus*. However, *D. viscosa*, *A. halimus* and *Z. fabago* could be compared only with wheat straw and switchgrass, and *P. miliaceum* with poplar or willow. This is particularly relevant for arid or semiarid areas where the cultivation of poplar and willow, or even *Miscanthus*, is restricted due to their high water requirement. The low values of *B. bituminosa* and *F. vulgare* make them inadequate for energy production by combustion.

However, the energy recovered from combustion depends on the moisture content of the biomass. Here, the maximum energy from combustion was calculated from the lower heating value. However, the plant biomass at harvest in the field needs to be dry for efficient combustion. The moisture content of the biomass at harvest ranged from 38 to 77% in *F. vulgare* and *Z. fabago*, respectively. Therefore, if the moisture content of the plant biomass is considered for the calculation of the potential energy (taking into account

the heat of condensation of water; dashed bars in Figure 2), then the potential energy recoverable from combustion decreases for all biomasses. In this case, the cultivation of *P. australis* and *A. donax* for biogas production would be more energetically efficient than for combustion, and species such as *Z. fabago* and *B. bituminosa* could not be considered adequate for combustion. Therefore, the moisture content of the plant biomass at harvest is very relevant for energetic and economic balances [11].

The drying of the biomass before combustion may imply an excessive cost that may make the process economically unviable. Therefore, special attention should be paid to the time of harvest; preferably, it should be at plant senescence, when the tissues are drying out, for biomass combustion, but fresh biomass can be used for biogas production through anaerobic digestion.

## 4. Conclusions

The concentrations of TEs in the plants from former mining sites did not affect biogas production, or the production of thermal energy according to the higher heating values. In particular, the species *P. miliaceum*, *P. australis* and *A. donax* have the most potential for the production of biogas. Contrastingly, *D. viscosa* and *A. halimus* have characteristics more adequate for direct combustion, due to their low ash and higher lignin concentrations, than for anaerobic digestion. However, the presence of elevated alkaline metal concentrations in their biomass may limit the usefulness of *A. halimus* and *Z. fabago* for combustion, restricting their use as bioenergy crops to non-saline sites.

The BMP values indicate that these species could be used as energy crops in arid or semiarid areas with results comparable with those of well-known energy crops such as *Miscanthus*, switchgrass, maize, milk thistle and sorghum silage. The highly productive *P. australis* and *A. donax*, with about 300 GJ ha$^{-1}$ y$^{-1}$, were the species with the highest potential annual energy production from biogas, followed by *P. miliaceum*.

In terms of combustion, *P. australis* and *A. donax* were the most efficient regarding the annual energy yield, followed by *P. miliaceum*; *B. bituminosa* and *F. vulgare* were the species least suitable for combustion. However, the moisture content of the plant biomass at harvest was shown to be a key factor for an efficient energetic balance, and the time of harvest needs to be optimized in order to control the moisture status of the biomass.

Therefore, among the species tested, *A. donax*, *P. australis* and *P. miliaceum* can be considered those most suitable for use as energy crops in the phytomanagement of TE-contaminated soils under Mediterranean conditions. However, other factors—such as the invasive character of *A. donax* and *P. australis*—should be taken into account when designing the corresponding remediation program. Considering all these aspects, *P. miliaceum* can be proposed as the most promising plant species for bioenergy production in soil phytostabilization.

**Supplementary Materials:** The following are available online at https://www.mdpi.com/article/10.3390/agronomy11091750/s1, Table S1: Macronutrient and sodium concentrations in the plants used in the experiment, Table S2: Pearson's coefficients for the correlations between the concentrations of the different TEs in the plants.

**Author Contributions:** Conceptualization, M.P.B. and R.C.; methodology, M.P.B. and R.C.; formal analysis, D.G., R.C. and M.P.B.; investigation, D.G., R.C. and M.P.B.; data curation, R.C. and M.P.B.; writing—original draft preparation, D.G., and M.P.B.; writing—review and editing, R.C. and M.P.B.; supervision, R.C. and M.P.B.; funding acquisition, R.C. and M.P.B. All authors have read and agreed to the published version of the manuscript.

**Funding:** This research was funded by the Spanish Ministerio de Ciencia e Innovación (MCI), the Spanish Agencia Estatal de Investigación (AEI) and the European Regional Development Fund (FEDER) through the project RTI2018-100819-BI00.

**Institutional Review Board Statement:** Not applicable.

**Informed Consent Statement:** Not applicable.

**Data Availability Statement:** The data presented in this study are available on request from the corresponding author.

**Acknowledgments:** The authors would like to thank David J. Walker for the English revision of the manuscript.

**Conflicts of Interest:** The authors declare no conflict of interest. The funders had no role in the design of the study; in the collection, analyses, or interpretation of data; in the writing of the manuscript, or in the decision to publish the results.

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
