# Peer review of "Potential of the Biomass of Plants Grown in Trace Element-Contaminated Soils under Mediterranean Climatic Conditions for Bioenergy Production"

_agronomy, doi:10.3390/agronomy11091750_

Round 1

Reviewer 1 Report

The manuscript entitles Potential for bioenergy production from the biomass of plants grown in trace‐elements‐contaminated soils under Mediterranean climatic conditions presents an original study, very well developed.

   I have only two remarks:

  • Page 332-333 - Is very difficult to read these graphs. Can you transform them in line chart with intervals? These lines might be rendered as shading to show the most common data values.
  • Page 456 – 457 - For a better and easy understanding, please put the entire name of the species in the graph.

Reviewer 2 Report

# manuscript: agronomy-1341728

# peer-review version: v1

# comments version: c1

# ---------------------------------------

# Title:

>> Title appear to be a bit twisted and can be restructured to make it easily palatable.

# ---------------------------------------

# Abstract:

>> Good structure and content. Basically, covers everything.

# ---------------------------------------

# Introduction

# # General comments

>> Nice framed to tell the story and covers good details and information.

>> Bulky and a bit long, but structured well.

# # Specific comments

>> Line 31: consider removing “the area of”

>> Line 50: “To date” is principally wrong, consider replacing

>> Line 83: Please avoid sentence start with “Then” throughout the text

>> Line 98: “We hypothesized that the …” is conceptually wrong when used after the experimental process and results. It might be better to use interpretation or similar here.

>> Line 98-102: Please restructure, based in above change.

# ---------------------------------------

# Materials and Methods

# # General comments

>> Clear and descriptive

>> Good enough information which can be used to reproduce the study, at least analyses part.

# # Specific comments

>> Line 160-161: Not clear information what is the purpose of this sentence? If the biogas reached its maximum, fundamentally AD process should not be stopped and prolonging the experiment will/can result in higher degree of degradation, overall higher biogas yield, result in lower residual methane production and lower emission. Should be restructured.

>> Line 171, : Use of units need consideration according to journal guidelines, however, for me it appeared to be wrong (mL g-1 VS) and a mix of mL/gVS and mL g-1 VS-1. Please consider correction throughout the text.

>> Miscanthus is a genus, so please italicize it throughout.

# ---------------------------------------

# Results and Discussion

>> Line 253-255: Highest is correct but other values are not similar. Rewrite the sentence.

>> Line 266: The word similar should carefully be used representing the ranges or heterogeneous data. Please consider replacing it.

>> Line 281: Consider using Bomb Calorimeter

>> Line 281, 284: Please change the “values of to” something appropriate or “values according to Reference [11, 49]”

>> Line 367: Consider rephrasing “ most of the rest of the species” to something more grammatically meaningful.

>> Line 424: “needed as part of the active site of enzymes” not correct, metal ions are needed for enzymatic activity or structural conformation but not for active site as it is described. Please rephrase.

>> Please correct unit annotation GJ ha‐1 yr.

# ---------------------------------------

# Figures

>> Figure 1: Figure are not good enough in the clarity and information they provide.

  • The points are very dense with borders and shape is absolutely not visible.

  • The error bars looks like percentage deviation and not providing any useful information, can be removed.

>> Figure 2: Energy unit mentioned in text and figure are not same.

# ---------------------------------------

# Tables

>> Table 1-4: The alphabetical letters are confusing and do not provide any clarity to the table, since they are not represented in table legend. Consider removing and represent better to make it easier to understand.

# ---------------------------------------

# Data availability

>> There cannot be any just reason not to include the raw/metadata with the manuscript. The statement regarding data could be provided later is absolutely/strongly discouraged. The statistical analyses/results are described in good details, however, in the absence of raw data, it is absolutely impossible for someone to verify and reproduce the results. The data presented in the manuscript must be readily available with the manuscript and is a good ethical practice.

# ---------------------------------------

# References

>> Might need to provide the accessible address for the reference 23-25.

# ---------------------------------------

# Complete manuscript

>> Good study and well presented in text. Some details like figures, tables however need to be improved. Submission of raw/metadata are highly encouraged.

Author Response

Please, see attached file.
